# Dilated Cardiomyopathy Associated with Paraquat Herbicide Poisoning

**Jaffer Ahmad [1], Kyla D'Angelo [2], Madalyn Rivas [3], Manpreet Mahal [3], Vikas Nookala [3], Dovile Kulakauskiene [1] and Amgad N. Makaryus [2,4,*]**

1. Department of Medicine, Nassau University Medical Center, East Meadow, NY 11554, USA; jahmad1@numc.edu (J.A.); Dkulakau@numc.edu (D.K.)
2. Department of Cardiology, Nassau University Medical Center, East Meadow, NY 11554, USA; kdangelo@numc.edu
3. Clinical Sciences, American University of the Caribbean School of Medicine, Pembroke Pines, FL 33027, USA; madalynrivas@students.aucmed.edu (M.R.); Manpreetmahal@students.aucmed.edu (M.M.); Vikasnookala@students.aucmed.edu (V.N.)
4. Donald and Barbara Zucker School of Medicine at Hofstra/Northwell, Hempstead, NY 11550, USA
* Correspondence: amakaryu@numc.edu; Tel.: +1-516-296-2567

**Abstract:** Dilated cardiomyopathy is a subset of cardiomyopathies defined by reduced ejection fraction of less than 45% and a dilated left ventricle. While dilated cardiomyopathy is common, its etiology is not always readily evident. Paraquat is used as an herbicide worldwide and is one of the main causes of fatal poisoning in underdeveloped countries in Asia, Central America, and the Pacific Islands. The most commonly affected organs are the lungs and kidneys. However, experimental research has shown that Paraquat can affect the heart indirectly through increased vascular permeability. In vivo animal studies have shown that paraquat poisoning causes myocardial contractile dysfunction by decreased fractional shortening and cardiac remodeling. We report the first case in published literature of a 52-year-old Hispanic man with dilated cardiomyopathy strongly associated with Paraquat exposure. It is important to obtain detailed medical history and proper diagnostic work-up including work, social, and family history, and echocardiography, baseline EKG, lab work, and ischemia cardiac testing as it can lead to improved diagnostic evaluation of possible etiologies of the commonly seen dilated cardiomyopathies and help identify less well-known etiologies as seen in our patient.

**Keywords:** dilated cardiomyopathy; paraquat; heart failure; cardiac CT angiography





## 1. Introduction

Dilated cardiomyopathy (DCM) is a subset of cardiomyopathies defined by a reduced ejection fraction less than 45% and a dilated left ventricle [1]. DCM is currently the most common subtype of cardiomyopathies, afflicting one in every 2500 adults from ages 20–50. Prevalence rates are significantly higher in men. Some causes of DCM are genetics, diabetes, obesity, hypertension, substance abuse including alcohol and cocaine, arrhythmias, certain chemotherapy drugs, coronary ischemia, valvular abnormalities, and toxin exposure [2,3]. Despite this, the etiology of DCM is not always readily evident and requires further investigation through thorough history, clinical examination, imaging, and genetic studies. Certain environmental factors can also play a contributing role in the development of DCM [4]. We present a case of DCM in an agricultural worker who recently immigrated to the USA from El Salvador.

## 2. Case Summary

A 52-year-old man with no past medical history presented to the emergency department complaining of abdominal pain, nausea and two episodes of non-bloody, nonbilious,

vomiting, and a chronic cough. His symptoms were treated with ondansetron, famotidine, and intravenous normal saline drip for hydration. Results of an abdominal ultrasound showed a hepatic cyst measuring $3.3 \times 3.2 \times 2.4$ cm, which was thought to be the cause of the patient's abdominal pain. Incidental findings of the obtained chest X-ray and abdominal CT revealed cardiomegaly and trace pericardial effusion. Routine laboratory studies showed the patient was hypokalemic, hypomagnesemic, and hyponatremic with mildly elevated hepatic enzymes and elevated creatinine in the setting of acute kidney injury. Urine analysis was normal.

The patient was admitted to the hospital for further treatment and investigation of the cardiomegaly. On further history, the patient stated that he had been recently seen at an urgent care for nausea and abdominal pain. Furthermore, the patient denied any alcohol, tobacco, or recreational drug use. He also denied any past medical history as he does not follow up with a primary care doctor. The patient furthermore denied any family history of heart failure, DCM, or aborted/resuscitated sudden cardiac death. The patient was active and could perform activities of daily living with no complaints and denied any symptoms of heart failure on admission or any time prior. In the past, he worked in agriculture spraying crops with an herbicide called paraquat for 12 years in El Salvador. Though this agent is notably toxic, the patient stated that he did not use personal protective equipment while handling the agent for all those years.

The patient's EKG (Figure 1) on admission demonstrated a left bundle branch block pattern not previously documented in the patient's medical history. Additionally, chest radiography (Figure 2) and computed tomography (Figure 3) on admission demonstrated cardiomegaly. Transthoracic echocardiography was technically difficult but demonstrated severe left ventricular systolic dysfunction with an ejection fraction of <15%. Initial brain natriuretic peptide (BNP) on admission was 981 pg/mL. The patient was recommended to start on 25 mg metoprolol succinate, 81 mg aspirin, 5 mg Lisinopril, and 40 mg atorvastatin. Metoprolol and Lisinopril were started initially as therapy for newly diagnosed DCM whereas aspirin and atorvastatin were started before coronary artery disease was ruled out with cardiac CT angiography (Figures 4 and 5). The patient developed electrolyte abnormalities after the first day of treatment, including hypomagnesemia and hyperkalemia, likely secondary to longstanding untreated fluid overload due to heart failure. In response to this and the noted acute kidney injury, Lisinopril was discontinued. Additionally, the patient's ALT and AST levels rose dramatically. Gastroenterology evaluation was obtained due to elevated liver function tests with a differential diagnosis of shock liver vs. Budd Chiari syndrome; therefore, N-Acetyl cysteine was prophylactically used and improved liver function over the subsequent week and atorvastatin was discontinued. The patient was continued on heart failure management. A repeat echocardiogram showed an improved ejection fraction of 25% with left ventricular dilation (left ventricular internal dimension in diastole 5.65 cm; left ventricular internal dimension in systole 4.95 cm; relative wall thickness 0.34; left ventricular mass 218.2 g; left ventricular mass index 131.8 g/m$^2$) and no left ventricular thrombus noted. Additionally, mildly reduced right ventricular systolic function was noted.

Over the course of the admission, the patient's presenting symptoms resolved. The patient was ambulatory and had no shortness of breath. He was discharged with home medications and a follow up appointment with cardiology within two days of discharge. He continues to do well on medical management for cardiomyopathy including sacubitril/valsartan after hyperkalemia was resolved.

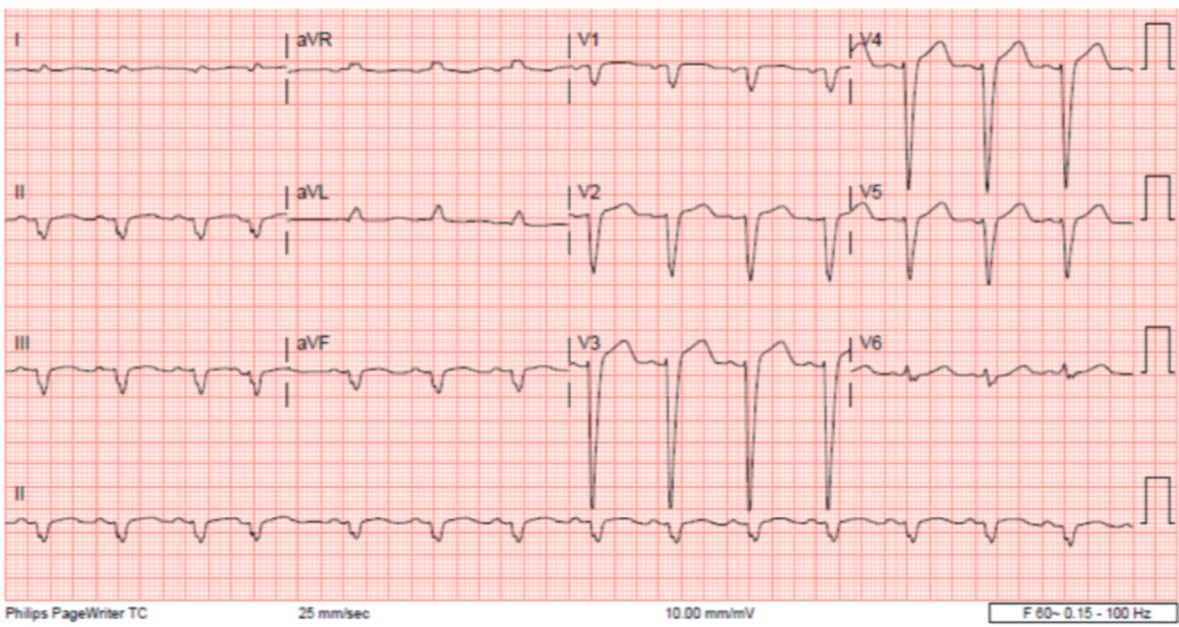

**Figure 1.** The patient's EKG on admission, demonstrating left ventricular hypertrophy not previously documented in the patient's medical history.

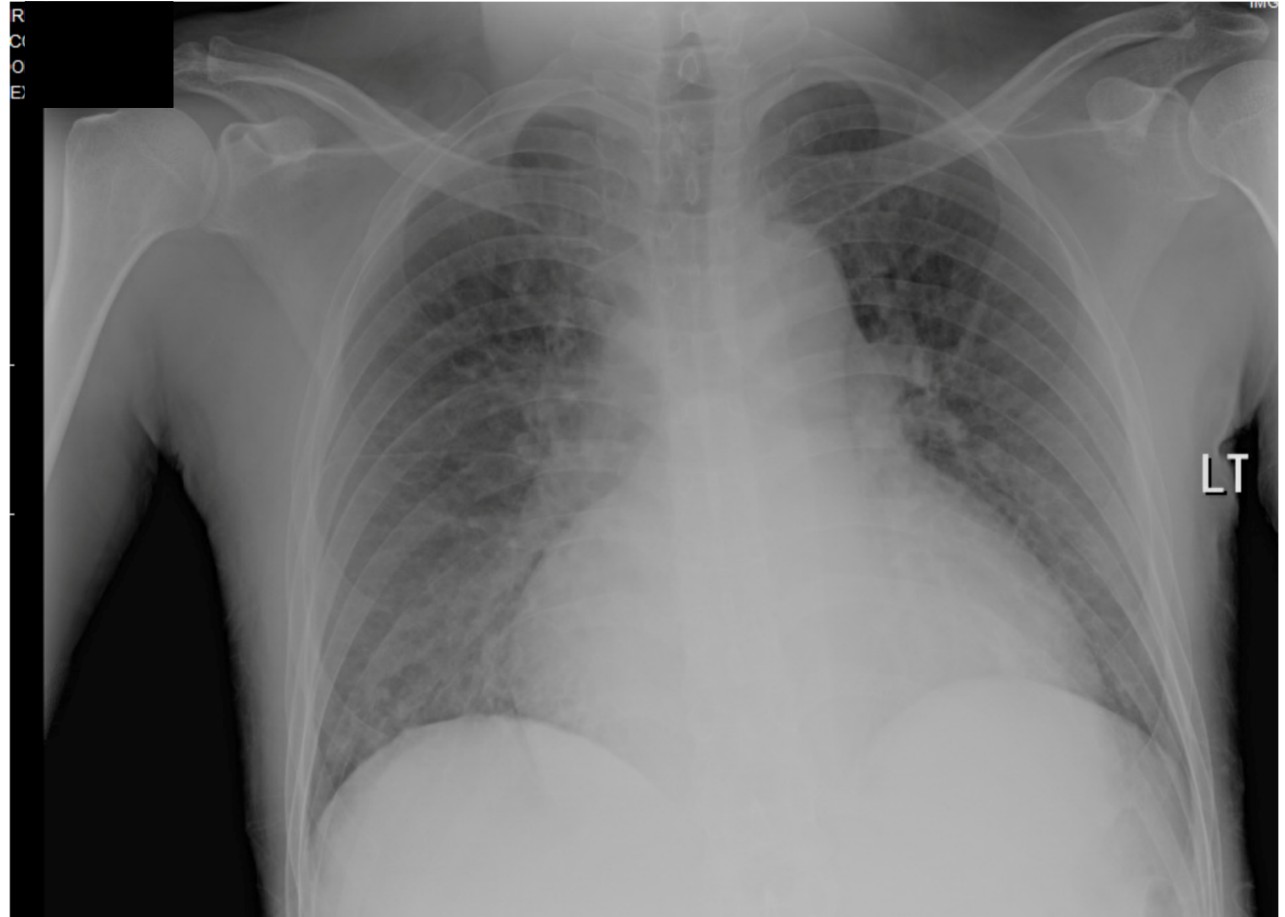

**Figure 2.** Chest radiograph on admission, demonstrating cardiomegaly.

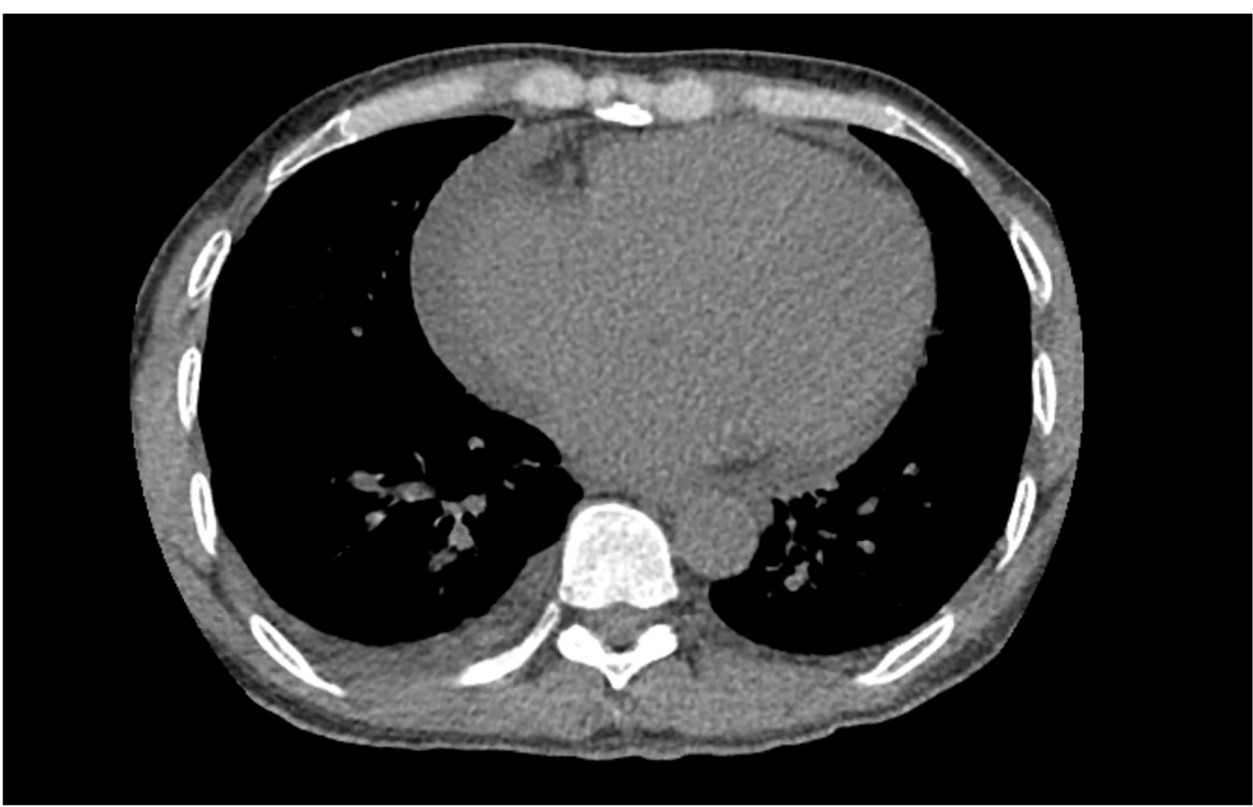

**Figure 3.** Segment of computed tomography (CT) of the thorax demonstrating cardiomegaly.

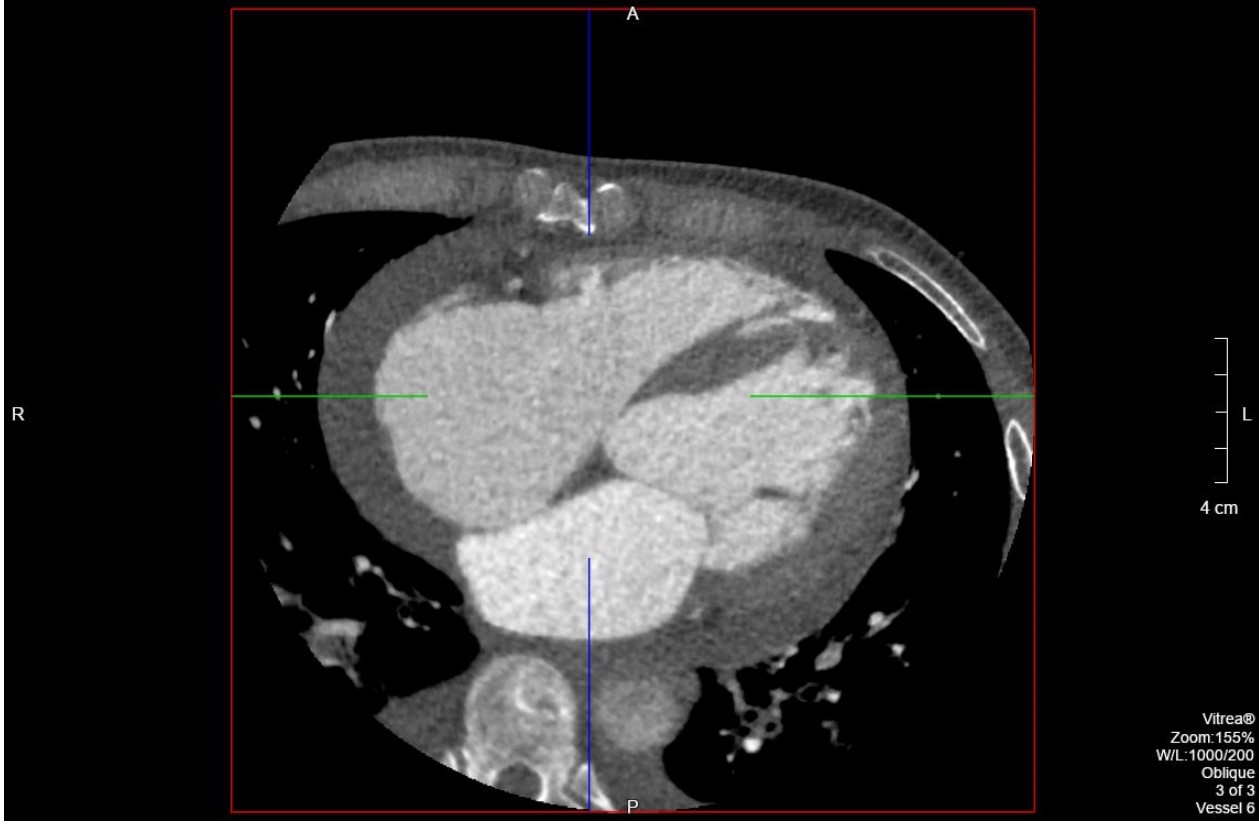

**Figure 4.** Segment of cardiac CT angiography demonstrating cardiomegaly and no coronary artery disease.

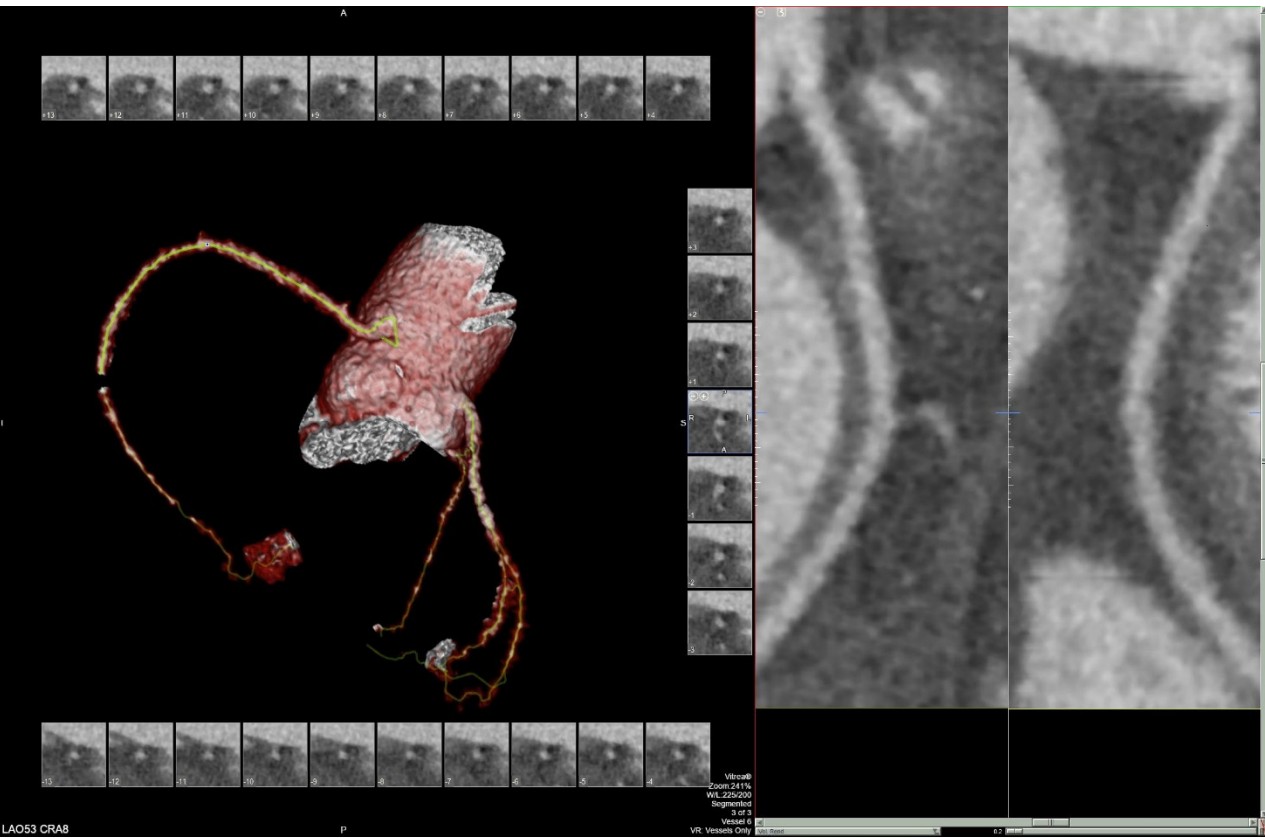

**Figure 5.** Segment of cardiac CT angiography demonstrating cardiomegaly and no coronary artery disease.

## 3. Discussion

Paraquat is a toxic herbicide commonly used in developing countries for broadleaf weed control. It is a fast-acting pro-oxidant compound that induces damage due to reactive oxidative species (ROS) development and subsequent stress injury via lipid peroxidation. It is also hypothesized that ROS act as second messengers and activate pro-inflammatory pathways, such as nuclear factor kappa-light-chains (NFkB), c-Jun N-terminal kinase (JNK), extracellular receptor kinase (ERK), and p38 microtubule associated protein kinase (MAPK). Activation of these pathways ultimately contributes to the organ damage seen in paraquat toxicity. Paraquat has been shown to cause multi-organ failure, most commonly affecting the pulmonary, renal, neurological, cardiovascular, and gastrointestinal systems via ROS accumulation and oxidative stress through mitochondrial injury and autophagy compromise [5]. Since paraquat is eliminated through the kidneys, damage to the kidneys by paraquat may decrease its elimination and cause a buildup of the toxin in tissues [6].

Several cases have illustrated the acute effects of paraquat toxicity in humans with the lungs being the most commonly affected. Damage to the lungs is the most lethal and least treatable manifestation of paraquat toxicity [7]. Paraquat toxicity has shown to have direct and indirect effects on the cardiovascular system. Direct cardiac injury has shown significant contractile dysfunction in rodents, as evidenced by decreased fractional shortening and cardiac remodeling (increased left ventricular end systolic diameter and end diastolic diameter). Histopathological studies showed that edema, congestion, and hemorrhage occurred in the myocardium [5]. Likewise, indirect effects of paraquat are caused by ROS leading to ischemic changes in the heart. Oxidative changes in the myocardium can rapidly change the structure and function of the cardiovascular system leading to various cardiovascular diseases. Rodent models of ventricular myocytes illustrated paraquat induced myocardial dysfunction may be attributed to altered mechanisms of calcium transport, leading to oxidative stress and myocardial dysfunction [8]. Research has indicated the

acute effects of paraquat toxicity on the cardiovascular system, but our patient presented with signs of chronic effects of the herbicide after 12 years of intermittent exposure.

Diagnosis in our patient was made after ruling out all other etiologies of cardiomyopathy through imaging and patient history. The patient denied any past or current heavy alcohol or drug use, ruling out substance abuse as a cause. He denied any past medical history such as hypertension, diabetes mellitus, or treatment with cardiotoxic drugs. He also denied history of viral or parasitic infections such as Chagas disease due to lack of symptoms such as malaise, fevers, weight loss or myocarditis, nor did he have any symptoms of such in the past. Sarcoidosis was also ruled out as the patient did not fit the clinical profile. He showed dyspnea on exertion, however he did not have any malaise, fatigue, fevers, wheezing, or abnormal weight loss, nor were there any cutaneous findings seen in sarcoidosis or ocular abnormalities. The chest radiograph of the patient showed no hilar adenopathy and was more consistent with a DCM picture. Regarding ischemic cardiomyopathy, CT angiography did not reveal any coronary plaques in the coronary arteries, ruling out atherosclerosis as a cause. Despite cardiac catheterization being the gold standard for ruling out atherosclerosis, cardiac CT angiography has a negative predictive value of 99% and our patient was very low risk for cardiac ischemia, making a cardiac catheterization an unnecessary invasive procedure.

Our patient further endorsed that he often used only his shirt or bandana to cover his face and wore regular clothing when handling paraquat instead of the recommended equipment. This resulted in his clothing becoming soiled with paraquat which he continued to wear throughout the day. Our patient stated he handled the herbicide biannually which included two harvest seasons. Each harvest season lasted at least two months in which paraquat was sprayed three times a week. On windy days, the team would attempt to avoid spraying against the direction of the wind in order to prevent inhalation. Since our patient has an extensive history of paraquat exposure with inadequate use of personal protective equipment (PPE), we believe that our patient's cardiomyopathy may be associated with paraquat toxicity.

There are no long-term studies describing the effects of paraquat in humans; however, several chronic effects for humans have been inferred from animal studies including Parkinson's disease and irreversible lung damage [1]. We believe that our patient likely developed his cardiomyopathy via chronic dermal exposure to paraquat. With dermal exposures occurring rarely and more acutely, we hypothesize that our patient could have potentially been exposed dermally via his soiled clothing which might explain the lack of respiratory symptoms. Our hypothesis may be corroborated by a study performed in Malaysian paddy farms which claimed that the type of spraying equipment, weather conditions, and use of PPE were the most important factors affecting risk of inhalation and dermal exposures [9]. We believe that the patient may have had minor dermal absorption via a smaller concentration of paraquat that may reflect why he did not have a fulminant reaction to paraquat [10]. Therefore, we believe that with ongoing chronic dermal exposure, he may have had repeated oxidative damage to his heart resulting in cardiomyopathy and heart failure.

Paraquat directly causes cardiac injury. Studies in vivo have shown that paraquat can cause significant contractile dysfunction in rodents by decreased fractional shortened cardiac remodeling (i.e., increased left ventricular end-systolic diameter and end-diastolic diameter). Histopathological studies also showed that edema, congestion, and hemorrhage occurred in the myocardium after administration of paraquat. These in vivo observations are consistent with the in vitro findings showing decreased peak shortening and max velocity of shortening and prolonged duration of lengthening. The above findings indicate that paraquat could compromise myocardial and cardiomyocyte contractile function. Furthermore, acute paraquat toxicity can also interrupt intracellular calcium homeostasis. This is evident by depressed peak and electrically stimulated release of intracellular calcium levels as well as a prolonged intracellular calcium clearance time [5].

In order to make a more definitive diagnosis of DCM secondary to paraquat exposure and not to idiopathic DCM, an invasive biopsy of the cardiac tissue and genetic testing to rule out other causes of DCM would need to be performed. While a cardiac biopsy would have been a reasonable approach for increased etiologic diagnostic utility in our case, it was not performed due to the invasive nature of the biopsy and the patient's hesitancy to proceed with the biopsy, especially noting his improvement with medical therapy. The evidence gathered through laboratory study, imaging, and patient history point to an association between paraquat exposure and this patient's DCM. However, after ruling out the other primary causes of DCM, namely presence of coronary artery disease, substance abuse including alcohol, diabetes, hypertension, and exposure to chemotherapy agents, it is likely that the patient's presentation and past exposure to paraquat is more than a simple association but likely points to a causal relationship.

## 4. Conclusions

We report the first documented case of dilated cardiomyopathy strongly associated to paraquat herbicide exposure. Dilated cardiomyopathy is prevalent worldwide and thus it is vital for clinicians to have a better understanding of the various etiologies, including newfound cases as ours of the aforementioned toxin, paraquat [2]. From this case, we hope to emphasize the significance of the side effects of paraquat in order to make clinicians aware of recognizing its manifestations. This would allow clinicians to provide early detection and effective treatment to patients.

**Author Contributions:** J.A.—principal author; K.D.—editing; M.R.—research and discussion section; M.M.—research and discussion section; V.N.—research and discussion section; D.K.—editing; A.N.M.—editing and final revision. All authors have read and agreed to the published version of the manuscript.

**Funding:** This case report was not funded by any organization, grant or contract.

**Informed Consent Statement:** All denoted images were obtained from the patient's medical records with the patient's and facility's permission. All images were made HIPAA compliant for publication and education use.

**Acknowledgments:** We would like to thank the medical students who worked with us tirelessly in their research on topics pertaining to this case, and Makaryus for his leadership, tutelage, and assistance acquiring the cardiac CT imaging for this patient.

**Conflicts of Interest:** This case report and its authors have no conflicts of interest to disclose and have no industry affiliations and there is no ethical dilemma in the study, treatment, or discussion of this case.

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
