# Peer review of "Dilated Cardiomyopathy Associated with Paraquat Herbicide Poisoning"

_clinpract, doi:10.3390/clinpract11030083_

Round 1

Reviewer 1 Report

Review

Title: Dilated Cardiomyopathy Associated with Paraquat Herbicide Poisoning

Journal: Clinics and Practice

Abstract

Properly written, however the last sentence is too general, and the text could be more concise. A sentence about diagnostic work-up is missing.

Introduction

The sentences “Some causes of DCM are genetics, diabetes, obesity, hypertension, substance abuse including alcohol and cocaine, arrhythmias, certain chemotherapy drugs, and toxin exposure (2,3). Of these causes, diabetes, obesity,  hypertension, and alcohol abuse are the most common.” are redundant, ischemic and valvular etiology is not mentioned. Or the word “primary” has to be used.

Examination – clinical examination.

At times genetic studies?

Case summary

Transaminitis?

Aborted/resuscitated sudden cardiac death

Please use only simple past tense.

The ECG is an LBBB pattern not LVH.

Poorly demonstrated?

What was the echocardiographic protocol, LV size, LV thrombus....other findings.

Which metoprolol (succinate?)

Was entertained?

Coronary ischemia?

Discussion

Is a little bit redundant, please reduce it, otherwise adequate.

Overall opinion

The case is worth to be published, however the text has to be improved, both gramatically and regarding its content.

Author Response

Reviewer 1:

Abstract

Properly written, however the last sentence is too general, and the text could be more concise. A sentence about diagnostic work-up is missing.

Inclusion of a statement regarding importance of detailed history and specific testing for diagnostic workup now included in the abstract.

Introduction

The sentences “Some causes of DCM are genetics, diabetes, obesity, hypertension, substance abuse including alcohol and cocaine, arrhythmias, certain chemotherapy drugs, and toxin exposure (2,3). Of these causes, diabetes, obesity,  hypertension, and alcohol abuse are the most common.” are redundant, ischemic and valvular etiology is not mentioned. Or the word “primary” has to be used.

Redundancies removed and ischemic and valvular etiology now included in the text

Examination – clinical examination.

Revision made in text.

At times genetic studies?

Mutations in certain genes can cause Familial DCM, further gene testing can aid in determining etiology. Testing can be warranted with a positive family history of DCM with little evidence towards any of the more common causes. This is noted in the statement mentioning genetic studies.

Case summary

Transaminitis?

Elevated hepatic enzymes highlighted in the history.

Aborted/resuscitated sudden cardiac death

Revision made in the text.

Please use only simple past tense.

Revision made in the text

The ECG is an LBBB pattern not LVH.

Revision made in the text and figure legend.

Poorly demonstrated?

Statement revised to state technical (sonographer) difficulty in obtaining images.

What was the echocardiographic protocol, LV size, LV thrombus....other findings.

Specific details of the echocardiogram have now been included.

Which metoprolol (succinate?)

Metoprolol Succinate administered, updated in the text.

Was entertained?

Sentence reworded in the text.

Coronary ischemia?

This has been reworded to denote presence of coronary artery disease (it does not rule out ischemia).

Discussion

Is a little bit redundant, please reduce it, otherwise adequate.

Due to the rarity of this etiology and the first presentation of DCM secondary to paraquat poisoning, our discussion section is extensive in order to provide adequate evidence in support of our claim. We believe it to be complete, but redundant areas have been adjusted.

Overall opinion

The case is worth to be published, however the text has to be improved, both grammatically and regarding its content.

Thank you for your input and support of our paper.

Reviewer 2 Report

The authors reported a patient who might develop dilated cardiomyopathy due to paraquat herbicide poisoning. Several concerns have been raised.

  1. How was the size of hepatic cyst?

  1. In Figure 1, the authors should present only ECG. This ECG really shows LVH instead of RVH?

  1. ACE inhibitor in general should be continued even though patients have mild hyperkalemia.

  1. Description of echocardiography data is strange.

  1. Did the authors use diuretics? Also, how about SGLT2 inhibitor and MRA?

  1. There is no citation of figures.

  1. There is no clinical data supporting the association between dilated cardiomyopathy and paraquat use.

Author Response

Reviewer 2:

The authors reported a patient who might develop dilated cardiomyopathy due to paraquat herbicide poisoning. Several concerns have been raised.

  1. How was the size of hepatic cyst?

The hepatic cyst measured 3.3x3.2x2.4 cm, the text now reflects the size.

  1. In Figure 1, the authors should present only ECG. This ECG really shows LVH instead of RVH?

We have edited the EKG description and figure legend reflecting left bundle branch block as opposed to LVH.

  1. ACE inhibitor in general should be continued even though patients have mild hyperkalemia.

ACE inhibitor was held not just due to hyperkalemia but also due to acute kidney injury as now noted in the text. 

  1. Description of echocardiography data is strange.

Echocardiography description has been updated in the text.

  1. Did the authors use diuretics? Also, how about SGLT2 inhibitor and MRA?

Diuretics were not used, although spironolactone was considered yet held due to labile blood pressure and hyperkalemia. SGLT2 inhibitors were not considered due to kidney disease being acute and reversible as opposed to chronic. MRAs were not used (although spironolactone was considered as noted above) due to acute kidney injury and labile blood pressure while on beta blockers.

  1. There is no citation of figures.

Figures are now cited in the paper text. Figure legends are noted below the work cited section of the document.

  1. There is no clinical data supporting the association between dilated cardiomyopathy and paraquat use.

Although we could not provide genetic testing or cardiac biopsy results, we are confident in the diagnosis and etiology due to the history and presentation of this patient. We are also confident in this diagnosis due to no other causes that could explain cardiomyopathy in this patient. This is highlighted throughout our discussion section.

Many thanks in advance for your consideration.

Round 2

Reviewer 2 Report

There are no further comments.